



# Optical and hygroscopic properties of black carbon influenced by particle microphysics at the top of anthropogenically polluted boundary layer

Shuo Ding[1], Dantong Liu[1], Delong Zhao[2,3], Kang Hu[1], Ping Tian[2,3], Fei Wang[2], Ruijie Li[2], Yichen Chen[2], Hui He[2], Mengyu Huang[2], Deping Ding[2]

[1]Department of Atmospheric Sciences, School of Earth Sciences, Zhejiang University, Hangzhou, China

[2]Beijing Weather Modification Office, Beijing, China

[3]Beijing Key Laboratory of Cloud, Precipitation and Atmospheric Water Resources, Beijing, China

Corresponding to: dantongliu@zju.edu.cn



## 1  Abstract

Aerosols at the top of planetary boundary layer (PBL) could modify its atmospheric dynamics by redistributing the solar radiation, and start to be activated to form low-level cloud at this layer. Black carbon (BC), as an aerosol component efficiently absorbing solar radiation, can introduce heating and positive radiative effects at this sensitive layer, especially in the polluted PBL over the continent. This study presents continuous measurements of detailed BC properties at a mountain site locating at the top of polluted PBL over the North China Plain, during seasons with contrast emission structure and meteorology. The pollution level was persistently influenced by local surface anthropogenic emission on daily basis through daytime convective mixing, but the concentration was also enhanced or diluted depending on air mass direction, defined as neutral, polluted and diluted PBL, respectively. Winter was observed to have a higher BC mass fraction (4-8%) than summer (2-7%). By resolving the detailed particle size-resolved mixing state of BC in optical and hygroscopic models, we found enhanced BC mass absorption cross section ($MAC_{BC}$) for polluted PBL (up to 13 $m^2g^{-1}$ at $\lambda=550nm$), and summer had a higher $MAC_{BC}$ than winter by 5%. The higher BC mass fraction in winter corresponded with a lower single-scattering albedo by 0.03- 0.09 than summer, especially the lowest for diluted winter PBL (0.86±0.02). The water supersaturation (SS) required to activate half number of BC decreased from 0.21±0.08% to 0.1±0.03% for winter diluted and polluted PBL; from 0.22±0.06% to 0.17±0.05% for summer. Notably, at the top of anthropogenically polluted PBL in both seasons, the enlarged BC with enhanced absorption capacity could be also efficiently droplet activated, e.g. winter (summer) BC with MAC of 9.84±1.2 (10.7±1) $m^2 \cdot g^{-1}$ could be half activated at SS=0.13±0.06% (0.18±0.05%). These BC at the top of the PBL can more directly interact with the free troposphere and be transported to a wider region, exerting important direct and indirect radiative impacts.





## 1. Introduction

Black carbon aerosol (BC) is strongly shortwave absorbing, wielding important climate warming impact in regional and global scale (Bond et al., 2013;Bond and Bergstrom, 2006). The emission of BC has large regional heterogeneity with higher impact over polluted regions (Ramanathan and Carmichael, 2008). The impacts due to BC heating are importantly determined by its vertical distribution in atmospheric column, which could lead to a more stable or convective planetary boundary layer (PBL) (Ding et al., 2016;Koch and Del Genio, 2010), e.g. an enhanced heating at higher level will depress the development of the layer below (Chung et al., 2002;Ramanathan et al., 2005;Hansen et al., 2005), while the heating can promote the convection above (Mcfarquhar and Wang, 2006;Rudich et al., 2003). Therefore, the properties of BC at the top of PBL are important, which may result in contrast impacts in perturbing atmospheric dynamics. In addition, surface emissions could serve regular cloud condensation nuclei (CCN) on daily basis through daytime convective mixing in the PBL (Bretherton and Wyant, 1997;Wood and Bretherton, 2004). Aerosols can be uplifted to the top of PBL and subsequently activated to incorporate into clouds. However, the way of BC from surface sources to be activated, after the PBL processing during vertical transport, is yet to be explicitly understood.

Though ground measurements of BC have been intensively conducted over the polluted North China Plain (NCP) region in last decades (Han et al., 2009;Cheng et al., 2011;Ji et al., 2018;Liu et al., 2019a), they were not able to represent the BC properties at the top of PBL. Additionally, recently conducted series of aircraft measurements provided comprehensive information about vertical distributions of BC over this region (Zhao et al., 2019;Ding et al., 2019a), but the manner of the aircraft measurement could not capture the variation of BC at certain level in sufficient time resolution. The nature of diurnal pattern of PBL means not only the surface concentration of pollutants is influenced by the daily evolution of PBL, but also the pollutants at the top of PBL will have diurnal variation which need stable measurements to be characterized. This is particularly the case if the top of PBL was continuously influenced by surface emissions. However such information is lack at the top of PBL over the polluted NCP region, and many models still reply on surface measurements to estimate the conditions at higher level (Guleria et al., 2014;Srivastava et al., 2012).

Microphysical properties of BC importantly determine its optical and hygroscopic properties. For example, the presence of coatings associated with refractory BC (rBC) may enhance its absorbing capacity (Liu et al., 2017), also enhances its hygroscopicity (Liu et al., 2013) and particle size, thereby modifying the potential to be droplet activated (Chuang et al., 2002;Panicker et al., 2016;Ding et al., 2019b). This study is for the first time to characterize the detailed BC microphysics at a mountain site located at the top of PBL, influenced by surface emission on daily basis over the NCP region. We investigated the optical and hygroscopic properties of BC at this level, as influenced by microphysical properties. Such information will support to constrain the impacts of BC in influencing the PBL dynamics and low-level cloud formation over this anthropogenically polluted region.

## 2. Site description, meteorology, cluster classification

Experiments in this study were performed in winter (Feb. - Mar.) and summer (Jun. – Jul.) of 2019 at a mountain site (115.78°E, 40.52°N, 1344 m), locating on the north of Taihang ridge to the northwest of central Beijing, shown in Fig 1. (a). The site is away from any local primary emissions, but the only sources of pollutants are contributed by lower-level surface emissions and reginal transport (the follow-up discussions).

Backward trajectories at the site are analyzed using the HYSPLIT 4.0 model (Draxler and Hess, 1998), for every 6-hour during the experimental period. The meteorological field uses the 1° × 1°, 3-hourly GDAS1



reanalysis product and trajectories back to 48h are calculated. The backward trajectories are used for further
cluster analysis, to group the backward trajectories with similar transport pathway, whereby the homogeneity
of trajectories is maximized in each cluster, meanwhile the heterogeneity among different clusters is
maximized (Makra et al., 2011). This is achieved by analyzing the spatial variance of each trajectory and the
total variance in each pre-defined cluster. The iteration next simultaneously calculates and assigns the
trajectories to the eventually merged clusters. This analysis has been widely used to identify the main transport
pathways of air mass (Markou and Kassomenos, 2010;Philipp, 2009;Jorba et al., 2004;Grivas et al., 2008),
and is performed using the built-in module in HYSPLIT model software.

## 3. Instrumentation

All aerosol measurements were performed behind a $PM_{2.5}$ impactor (BGI SCC1.829), and were dried by a
Nafion tube prior to the sampling of the instruments. BC was measured by a single-particle soot photometer
(SP2) (DMT Inc. USA). This instrument uses laser-induced technique to incandesce BC-containing particle
(Schwarz et al., 2006;Liu et al., 2010). The measured incandescence signal of individual BC particle can be
converted to a refractory BC (rBC) mass, which was calibrated using Aquadag® BC particle standard
(Acheson Inc.), and a factor of 0.75 was applied to correct for the ambient rBC mass (Laborde et al., 2012).
The core size ($D_c$) of BC is calculated from the measured rBC mass by assuming a material density of BC of
1.8 g/cm$^3$ (Bond and Bergstrom, 2006). The scattering cross section of BC is derived by the leading edge only
(LEO) method (Gao et al., 2007) on the measured scattering signal of individual BC particle. The entire
particle size of BC containing particle ($D_p$) including coatings is determined by matching the measured
scattering cross section with the modelled one using a Mie lookup table (Liu et al., 2014;Taylor et al., 2015).
The bulk coating thickness ($D_p/D_c$) in a given time window is calculated as the cubic root of the total volume
of BC-containing particle weighted by the total volume of rBC (Liu et al., 2014):
$$\frac{D_p}{D_c} = \sqrt[3]{\frac{\sum_i D_{p,i}^3}{\sum_i D_{c,i}^3}} \qquad (1),$$

where $D_{p,i}$ and $D_{c,i}$ are the diameters for the $i^{th}$ single particle, respectively. The count (or mass) median
diameter (CMD, or MMD) is derived from a number (or mass) size distribution, below and above which size
the number (or mass) concentration is equal. Modelled mass absorption cross section (MAC) of individual
BC-containing particle is calculated by applying the Mie theory (Bohren, 1998) with a core-shell mixing state
assumption. The bulk MAC for a given time window can then be determined by the total MAC of each BC
particle divided by the total rBC mass, expressed as:
$$MAC = \frac{\sum_i MAC_i \times m_{rBC,i}}{\sum_i m_{rBC,i}} \qquad (2),$$

where $MAC_i$ and $m_{rBC,i}$ is the MAC and rBC mass for each particle respectively. An example to calculate
the MAC from single particle information is shown in Fig. 5f. The absorption coefficient $k_{abs}$ (in Mm$^{-1}$) is
calculated as the MAC (m$^2$·g$^{-1}$) multiplying the rBC mass concentration (μg·m$^{-3}$) in each size bin, then
integrated throughout the size distribution:
$$k_{abs} = \sum_i MAC(D_{p,i}, D_{c,i}) m(logD_{c,i}) \Delta logD_{c,i} \qquad (3),$$

where $m(logD_{c,i})$ is the BC mass concentration at each $D_c$ bin.
the volume ratio between coating and rBC could be obtained from $D_c$ and $D_p$ of individual BC:
$$\frac{\varepsilon_{coating}}{\varepsilon_{rBC}} = \left(\frac{D_p}{D_c}\right)^3 - 1 \qquad (4),$$

where $\varepsilon_{coating}$ and $\varepsilon_{rBC}$ is the volume fraction within BC particle. The hygroscopicity parameter of BC



($\kappa_{BCc}$) could be calculated with known $\kappa_{coating}$ and $\kappa_{rBC}$=0, based on Zdanovskii–Stokes–Robinson (ZSR) rule
(Stokes and Robinson, 1966), expressed as,
$$\kappa_{BCc} = \varepsilon_{coating} \times \kappa_{coating} \qquad (5),$$
In this study, the main focus is the coating abundance distribution across all BC size distribution, but the
variation of coating composition is to a less importance, thus a constant $\kappa_{coating}$=0.3 is used to represent typical
environment containing aged aerosols (Pringle et al., 2010). An example to calculate $\kappa_{rBC}$ from single
particle information is shown in Fig. 5g.
Total aerosol size distribution was measured by a Scanning Mobility Particle Size (SMPS, TSI Inc. Model
3936) at mobility diameter 15-700 nm. The total particle mass below diameter 1 μm (PM$_1$) is derived from
the SMPS size distribution by assuming a mean particle density of 1.45 g cm$^{-3}$ (Liu et al., 2015;Cross and et
al., 2007). The scattering cross-section $\sigma_{sc}$ (μm$^2$) of particle at all sizes is calculated by applying Mie
calculation assuming a refractive index (RI) of 1.48+0i (Liu et al., 2009) (a positive imaginary RI considering
the BC mass fraction is tested to have a minor influence in total scattering within 3%), thereby calculating the
scattering coefficient $k_{sca}$ (in Mm$^{-1}$) of all aerosols:
$$k_{sca} = \sum_i \sigma_{sc}(D_i) n( logD_i)\Delta logD_i \quad (6),$$
where $n$ (logD$_i$) represents the particle number concentration at the i$^{th}$ size bin. Single-scattering albedo (SSA)
is derived as the quotient of scattering coefficient ($k_{sca}$) divided by extinction coefficient ($k_{abs}$+ $k_{sca}$).

## 4. Results and discussion

### 4.1 Classification of PBL

Clustered air masses for both seasons are shown in Fig. 2, and three clusters are classified for each season.
For both seasons, Cluster 1 (C1) represents the air masses from the most intensive anthropogenically
influenced regions according to the BC emission inventory (Fig. 1b). The seasonal difference is the winter
pollution was transported from the west over Shanxi and Hebei province, while was southerly in summer
mainly from the North China Plain over Hebei province. Cluster 2 (C2) passed over similar regions in both
seasons from the cleaner north. Cluster 3 (C3) both had a longer transport than other clusters, from northwest
and northeast in winter and summer respectively. The diurnal variations of PBL height (PBLH) corresponding
with periods of each cluster are shown in Fig. 2(d, h), with pronounced diurnal pattern. Apart from C3, winter
had a lower PBLH than summer, i.e. the mountain site was slightly above or below the well-developed PBL
top in winter and summer, respectively. The consistent diurnal variation of PBL means the mountain site was
persistently influenced by the surface sources through daytime convective mixing, when pollutants were
transported in the polluted PBL. However, the actual concentration will depend whether the site had been
contributed by additional sources or dilution (Fig. 1c).
Fig. 3 shows the temporal evolution of BC mass and PM for both seasons classified by air mass clusters,
with right panels showing the frequency under each cluster. Fig. 4 shows their diurnal variations. For all
experimental period, the concentration level of BC mass generally positively correlated with the PM mass,
both indicating the pollution levels. Among the three clusters, C2 had the medium mass concentration, with
over 85% frequency of the air masses contributed by local regions (±1º around the measurement site), and
additional air masses were from the cleaner northerly direction, similar for both seasons (Fig. 2). It is likely
that this cluster was mainly contributed by local emission, and occasionally diluted to some extent by the air
mass from less polluted region. Here this cluster is defined as "neutral PBL", to be discriminated with the
other two clusters. In winter C2 showed clear diurnal variation of BC and PM$_1$ concentration with significant





enhancement in the daytime (Fig. 4b and e) when fully developed PBL (Fig. 2d). This diurnal pattern on the
mountain was contrast with the usual surface measurement, when the daytime PBL development could dilute
the concentration (Liu et al., 2019a;Han et al., 2009). This opposite trend on the mountain site was consistent
with the fully developed PBLH at 12:00-16:00 (Fig. 2d) when the top of PBL reached the mountain site at this
time, and pollutants were transported through convective mixing from the surface sources (Fig. 1c). In
midnight, the nocturnal depressed PBL trapped the surface pollution towards mountain, and the subsiding
cleaner air in free troposphere may have diluted the concentration of pollutants on the mountain site (Sullivan
et al., 1998;Bennett et al., 2010). The enhanced concentration from midday in summer was also observed (on
the mean, Fig. 4a) but not as pronounced as winter, probably due to some wet removal during the daytime
vertical transport from the surface, given the high RH in the summer (Fig. S1).

158       Consistent with the combined back-trajectory and emission analysis above, C1 had the highest BC for both
seasons (1.0 ± 0.5 and 0.4±0.2 µg m$^{-3}$ for winter and summer, respectively) and PM mass (23.8 ± 10.3 and
13.4±9.5 µg m$^{-3}$). The concentration of BC mass was enhanced by a factor of 2.8 (1.7) higher than that in C2
for winter (summer), with winter having mass concentration frequently exceeding 1 µg m$^{-3}$. It clearly shows
the diminished diurnal variation compared to the neutral PBL of C2, but all increased throughout the day and
night (Fig. 4a and d), particularly for winter. This is because besides the persistent influence of daytime
convective mixing as the neutral cluster in C2, C3 cluster had additional contribution from wider polluted
regions, hereby defined as "polluted PBL" (Fig. 1c). This contribution by regional transport was not related
to a diurnal pattern. The less enhancement in summer suggested the overall lower surface emission of the
surrounding regions in warm season.

168       C3 had the lowest pollution level among all air mass clusters with a lower BC and PM mass than C2 by a
factor of 2-4. Similarly, low BC (0.09±0.03 µg m$^{-3}$) and PM$_1$ mass (1.8±0.4 µg m$^{-3}$) were for both seasons. C3
represented the air masses from regions with low emissions (Fig. 1b). In addition, the faster transport (shown
as the longest path for back-trajectories among clusters, Fig. 2c) and the highest PBLH (Fig. 2d) in C3 could
efficiently dilute the pollution in the PBL, hereby termed as "diluted PBL". Compared to C2, C3 can efficiently
disperse and reduce the pollutants being vertically transported to the mountain site from the surface, thus with
no apparent diurnal pattern of pollution concentrations neither (Fig. 4c).

175       BC mass fraction in PM$_1$ was overall higher in winter than summer, on average at 5.1±1.7% and 3.6±2.3%
respectively, with both seasons showing a frequency distribution skewed to higher values up to 10% (Fig. 3 k
and l). Note that in summer the BC mass fraction among clusters had no discernible differences with only C2
showing to be slightly higher. The overall higher BC mass fraction in winter may result from the seasonal
variation on emission structure, that the additional primary emissions from heating activities may have
introduced more fraction of BC (Chow et al., 2011;Liu et al., 2018). However the lowered BC mass fraction
for the polluted PBL in winter (with significant reduction for fraction >7%) may result from the enhanced
secondary formation under polluted environment (Volkamer et al., 2006;Hallquist et al., 2009). The
comparable BC mass fraction between C1 and C3 in summer (but all lower than neutral PBL) may result from
the dominant process of enhanced secondary formation (reducing PM$_1$) and particle scavenging (reducing BC),
for polluted and diluted PBL, respectively. There was a notably higher BC mass fraction at 6.7±1.5% for the
diluted PBL in winter, suggesting a much reduced secondary aerosols in less-polluted environment, and the
removal process of BC had been less efficient than other substances (Koch and Del Genio, 2010).

## 4.2  Particle microphysics of BC

190       BC core size showed no variation among all PBL types in both seasons. The mass (count) median diameter
of BC core is 203±12 nm (106±7 nm) and 167±11 nm (85±5 nm) for winter and summer respectively. The BC





core size could be used to discern emission sources, such as BC from solid fuel burning tended to have larger
MMD compared to those from traffic sources (Liu et al., 2014). The larger MMD in winter than summer was
consistent with previous observation over this region (Ding et al., 2019a;Hu et al., 2020), which may indicate
additional sources from residential solid fuel burning for heating in cold season.
The coatings relative to BC, as reflected by $D_p/D_c$, was positively correlated with the pollution level of the
PBL (represented by $PM_1$ mass loading) in winter (Fig. 5a), and in summer from neutral to polluted PBL (Fig.
5b). The polluted PBL in winter had the highest $D_p/D_c$ (1.76 ±0.3), in line with the most polluted condition.
For both neutral and diluted PBL, winter had higher BC coatings than summer. Apart from some periods in
summer, the diluted PBL occasionally showed higher $D_p/D_c$ up to 1.8, comparable with the high end of that
for polluted PBL.
The warmer temperature in summer may have caused a more likely tendency for some semi-volatile particle
species to evaporate, hereby reduced coatings on BC than in winter even at the same pollution level, which is
consistent with previous ground studies over this region (Liu et al., 2019a). The highest coatings were
associated with the most polluted condition, implying that the regional transport of air mass from wider
polluted region may have advected both primary emissions and gas-precursors, during when some gas
partitioning processes may have occurred on BC during transport and caused the high coatings. Under clean
environment, summer showed remarkably higher coatings than winter, which may result from a more intense
solar radiation received at the mountain site in summer, where photochemical reactions may have caused
significant formation of secondary particulate matter (Xu et al., 2017;Bao et al., 2018;Liu et al., 2019b).
The detailed size distributions of uncoated and coated BC are shown in Fig. 5 c-e. Here three typical
examples are chosen to represent the polluted and diluted PBL in winter (case 1 and case 2), and polluted PBL
in summer (case 3). Comparing between case 1 and case 2, it showed the polluted period had caused increased
size for all particle from peak diameter (d) less than d=30 nm to the accumulation mode of d=120 nm. Coated
BC diameter peaked from 150 to 220 nm. It means the condensation process had occurred on all particles and
enlarged their sizes including BC. The coated BC in most polluted PBL has significantly extended its coated
size up to d=400-700nm. Note that for the clean PBL in winter, the coated BC was populated at d=150nm,
leading to a higher BC number fraction of 22% at this size. For polluted PBL in summer, peaking diameter of
all particles (d=450nm) was lower than winter, and coated BC also peaked at a smaller size (d=153 nm). Fig.
S3 shows almost consistent coated BC sizes in summer (120-130nm), which was smaller than winter because
of the smaller core size (though a higher $D_p/D_c$).
Detailed core-size resolved mixing states, expressed in a space of coated versus uncoated BC diameter, are
shown in Fig. 5 f-h, corresponding to the three cases above. $D_p=D_c$ indicates no coatings on BC and coating
increases when $D_p$ is larger than $D_c$. For polluted PBL in winter, most BC population lied between 100-200
nm core diameter and 200-300 nm coated diameter, in contrast with the diluted PBL that a significant fraction
of uncoated BC and coated diameter at 100-140 nm. Summer polluted PBL showed reduced fraction of bare
BC but a range of coating thickness for coated BC. Such analysis in mapping the size-resolved mixing state
of single BC particle on the space of uncoated-coated size, can resolve the optical (contour in Fig. 5f) and
hygroscopic (contour in Fig. 5g) properties that BC particle microphysics could influence, which will be
discussed next.

## 4.3 Optical properties of BC

Fig. 6 a-b and Fig. S4 show the mass absorption cross section (MAC) of uncoated and coated BC for different
PBL types in both seasons. Summer showed systematically higher MAC (7.2±0.1 $m^2 \cdot g^{-1}$) than winter (6.6±0.3
$m^2 \cdot g^{-1}$) for uncoated BC, because the BC core was smaller in summer than winter (Fig. S2 and Fig. S3). The





MAC for coated BC was largely modulated by coatings, showing positive correlation with $PM_1$ in winter
($MAC_{BC,coated}=0.09\times PM_1+8.7$, r=0.79) and summer ($MAC_{BC,coated}=0.12\times PM_1+9.5$, r=0.5), corresponding with
the diluted to polluted PBL types. The enhanced absorption efficiency of coated BC with increased pollution
level was consistent with previous studies over this region (Zhang et al., 2018;Ding et al., 2019a). The mean
$MAC_{BC,coated}$ for polluted PBL in summer ($11\pm 1$ $m^2\cdot g^{-1}$) was higher than that in winter ($10.7\pm 0.9$ $m^2\cdot g^{-1}$) by
2%. The neutral and diluted PBL also had higher MAC in summer than winter (by 8% and 22% respectively).
The overall higher absorption efficiency of BC in summer (especially in diluted PBL) was due to the smaller
core size, leading to a higher baseline MAC for uncoated BC by 10% than winter. This effect prevailed the
even lower coating amount and associated absorption enhancement in summer (Fig. 5b). Notably, the $MAC_{BC}$
for diluted summer PBL had been enhanced by both reduced BC core size and increased coatings (Fig. 5b)
reaching almost equivalent MAC compared to winter neutral PBL. This means at the top of the clean PBL in
summer, there was a BC layer with high absorbing capacity (in contrast to winter with much lower MAC),
which may efficiently absorb the strong solar radiation.
The single-scattering albedo at $\lambda=550nm$ ($SSA_{550}$) in winter was systematically lower for all PBL types (Fig.
6 c-d and Fig. S4), lowered by 0.06, 0.05 and 0.08 than summer for diluted, neutral and polluted PBL,
respectively. The decreased $SSA_{550}$ was in line with the increased BC mass faction (Fig. 3), and also influenced
by the absorbing efficiency. Our winter results here were comparable to previous aircraft measurements over
Beijing region in cold season of 2016, with SSA ranging from 0.8 to 0.98 from clean to heavily polluted period
in the PBL (Tian et al., 2020). For the diluted PBL in winter, $SSA_{550}$ could reach as low as $0.86\pm 0.02$ when
BC mass fraction reached $7\pm 1.5\%$ (Fig. 3k). This was within the range of that study in clean period at
SSA=0.8-0.85 in the PBL, and the neutral and polluted PBL were within similar range in transition and fully
polluted PBL at SSA=0.85-0.98, suggesting the clusters here may represent different development stages of
pollution events. The lower SSA in winter PBL than summer tended to induce radiative effects towards more
positive effect (Hansen et al., 1981;Haywood and Ramaswamy, 1998) at the top of PBL, in particular for the
clean winter PBL.

## 262   4.4 Hygroscopic properties of BC

The droplet activation of BC is determined by the particle size and hygroscopicity, where the coatings play
roles in enlarging entire BC size (Dusek et al., 2006a;Dusek et al., 2006b) and increasing its hygroscopicity
(Liu et al., 2013). Both factors can be obtained by our observation of size-solved mixing state of BC. The
coated BC diameter based on number concentration (coated CMD) is shown in Fig. 7 a, b. The CMD of coated
BC in winter polluted PBL was $0.19\pm 0.05$ μm, which were the largest among PBL types. Generally, coated
BC CMD was in positive correlation with the PM, in winter fitted as $CMD=0.003\times PM_1+0.14$ (r=0.69) and in
summer as $CMD=0.001\times PM_1+0.11$ (r=0.73). The coated BC CMD in winter PBL was larger than summer by
14 nm, 35 nm and 56 nm for diluted, neutral and polluted PBL, respectively.
The hygroscopicity parameter of BC-containing particle ($\kappa_{BCc}$) at the size of CMD is positively correlated
with coating content. In winter $\kappa_{BCc}$ was at $0.15\pm 0.02$, $0.20\pm 0.03$ and $0.24\pm 0.03$ for diluted, neutral and
polluted PBL respectively, while the corresponding values were $0.19\pm 0.03$, $0.20\pm 0.03$ and $0.22\pm 0.03$ in
summer, consistent with the reduced $D_p/D_c$ from polluted PBL to diluted PBL for both seasons. $\kappa_{BCc}$ could be
fitted as $\kappa_{BCc}=0.003\times PM_1+0.17$ (r=0.77) and $\kappa_{BCc}=0.004\times PM_1+0.17$ (r=0.49) for winter and summer
respectively. Note that summer diluted PBL had significantly higher $\kappa$ ($0.19\pm 0.03$) than winter for the same
PBL type.
After obtaining the CMD of coated BC and the corresponding $\kappa_{BCc}$, a critical water superstation (SS) could
be derived from the Kölher model (Fig. S5) for BC at the size of CMD to be activated. By assuming that all
of the population larger than CMD could also be consequently activated (because of the larger particle size





and higher hygroscopicity), the SS obtained above is thus the lower estimate for half of the number population
of BC to be activated, termed as $SS_{half}$. In line with the increased coated BC size and particle hygroscopicity,
$SS_{half}$ decreased with increased pollution level, from 0.21±0.08% to 0.1±0.03% for winter diluted to polluted
PBL; from 0.22±0.06% to 0.17±0.05% for summer. This highlights the lowest possible SS required to activate
BC in polluted winter PBL, and for the same PBL type, summer will need a higher SS, apart from occasionally
some lower $SS_{half}$ for summer diluted PBL. This potential CCN ability of BC is derived from the physical
properties of BC itself, but the actual activation of BC depends on the ambient superstation condition which
is determined by the size distribution of existing droplets and other aerosols competing CCN (Pruppacher et
al., 1998;Mcfiggans et al., 2005). The results here generally consistent with a previous surface measurement
of BC CCN-activation in urban Nanjing as constrained by size-resolved compositions (Wu et al., 2019), when
an activation fraction of 33% at SS=0.1% was found. Previous study using flight measurements over NCP
region (Ding et al., 2019a) found a SS=0.08% required to activate half of the BC number in heavy pollution
condition, consistent with the polluted PBL here.

## 5. Conclusion

By performing continuous measurement on a mountain site located at the top of planetary boundary layer
(PBL) over the north China Plain region in winter and summer, the optical and hygroscopic properties of BC
were investigated. We identified three types of PBL, all persistently influenced by surface anthropogenic
emission on daily basis through daytime convective mixing, but could be either enhanced or diluted subject
to received air masses. By investigating the detailed microphysical properties of BC, this study provides a
clear picture of optical and hygroscopic characteristics of BC at the top of anthropogenically influenced PBL.
Highlighted information includes higher BC mass fractions in winter than summer, corresponding with a lower
single-scattering albedo by 0.05-0.08, especially the lowest for diluted winter PBL (0.86±0.02); both mass
absorbing efficiency and CCN ability of BC are positively correlated with the pollution level of PBL, due to
enhanced coating content under more polluted environment, e.g. from diluted to polluted PBL, coating content
increased by 39% (11%), absorbing efficiency increased by 31% (10%), and the water supersaturation in
activating half number of BC decreased by 53% (26%) for winter and summer respectively. It clearly
demonstrates that BC with higher coating content could be efficiently incorporated into liquid clouds, and
meanwhile these BC had high absorbing capacity, which means these highly-absorbing BC may have great
potentials for in-cloud heating (Jacobson, 2012;Nenes and et al., 2002).
Compared to surface measurement, the results here are more directly linked to the aerosol properties closer
to the condensation level which is subsequently CCN-activated. BC located at this layer more importantly
determines its heating impacts due to receiving a stronger solar radiation. Rather than being subject to
significant scavenging processes of low-level emissions, BC transported to the lower free troposphere may be
transported to a wider region (Weinzierl, 2008;Yang et al., 2018), exerting regional direct and indirect radiative
impacts.

## Acknowledgment

This research was supported by the National Key Research and Development Program of China
(2016YFA0602001), National Natural Science Foundation of China (41875167, 41675038).
There are no potential conflicts of interest.
Data for this study are available from the file sharing link (https://pan.baidu.com/s/1YqnLHZly24URgULZ-
HIlOw) using extracting code u457.



## Author Contributions

DL, DZ and DD led and designed the study. SD, DL, DZ, KH, PT, RL,YC, FW, KB, HH, and MH set up and conducted the experiment. SD, DL and KH contributed to the data process. SD, DL wrote the paper.

## Figures and captions

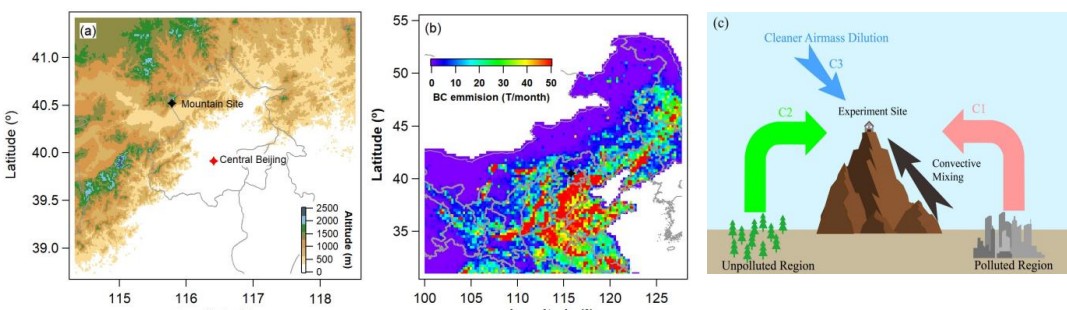

Fig. 1. Experimental site descriptions. (a) The location of the experimental site and central Being, marked with black and red star respectively, where the color bar denotes the terrian height. (b) the monthly BC emission inventory in China (Li et al., 2017). (c) Schematic illustration for different types of PBL defined in this study.

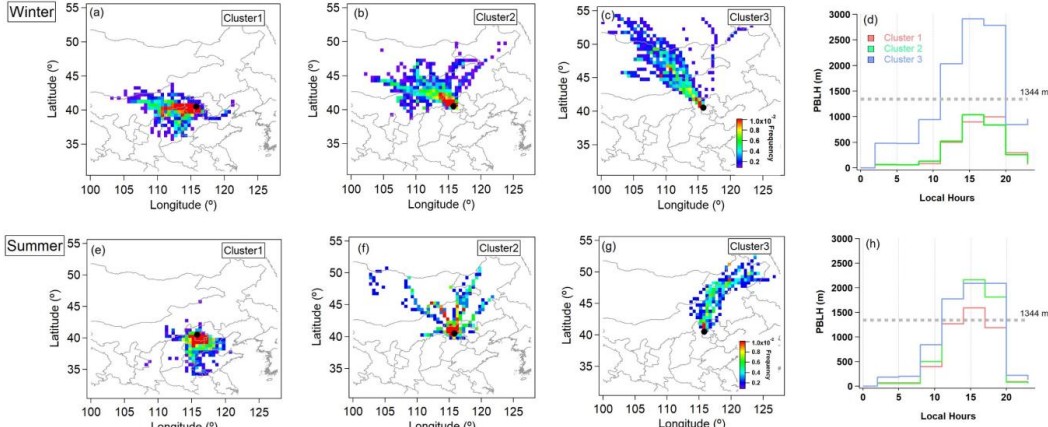

Fig. 2. The clustered backward trajectories from HYSPLIT model in both seasons: (a)-(c) for winter and (e)-(g) for summer, colored by occurrence frequency in each geographic grid. (d) and (h) are the diurnal variation of the height of PBL for the three clusters in both seasons, with the dashed line denoting the mountain site altitude.

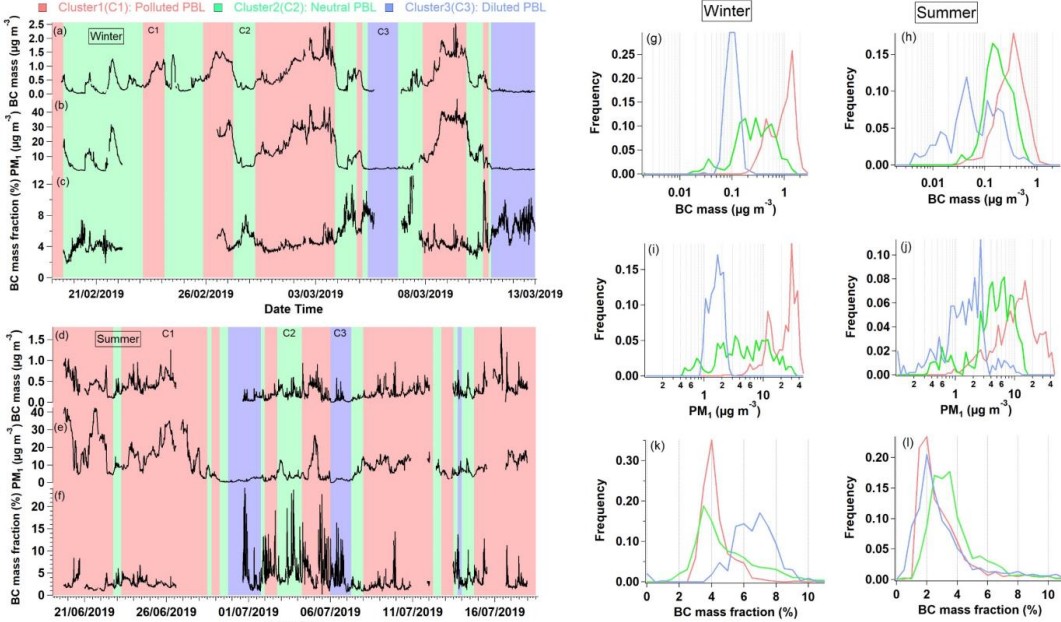

Fig. 3. Time series of BC mass (a)(d), PM$_1$ (b)(e), BC mass fraction (c)(f) at both seasons, shaded by the
periods in identified three clusters, by red, green and blue corresponding to cluster 1 (C1), cluster 2 (C2),
cluster 3 (C3) respectively. Frequency histograms of BC mass (g)(h), PM$_1$ (i)(j), and BC mass fraction (k)(l)
for each cluster in both seasons.



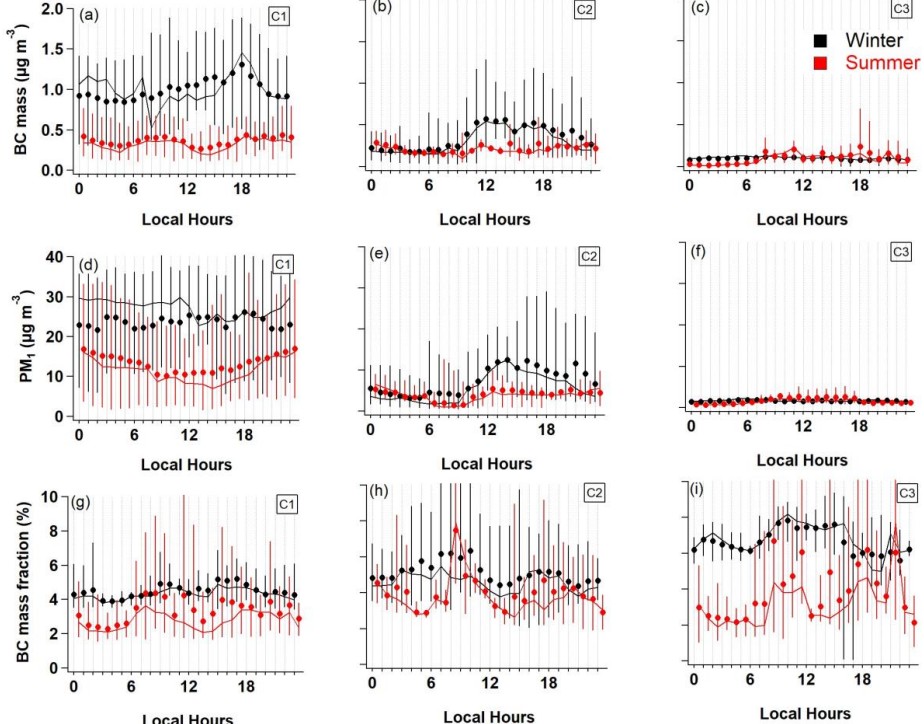

Fig. 4. Diurnal variarion of BC mass (a-c), $PM_1$ (d-f) and BC mass fraction (g-i) for the three PBL types in both seasons, where black and red denote the winter and summer respectively. The solid circles, lines and whiskers denote the mean, median, 25th, 75th percentile respectively.

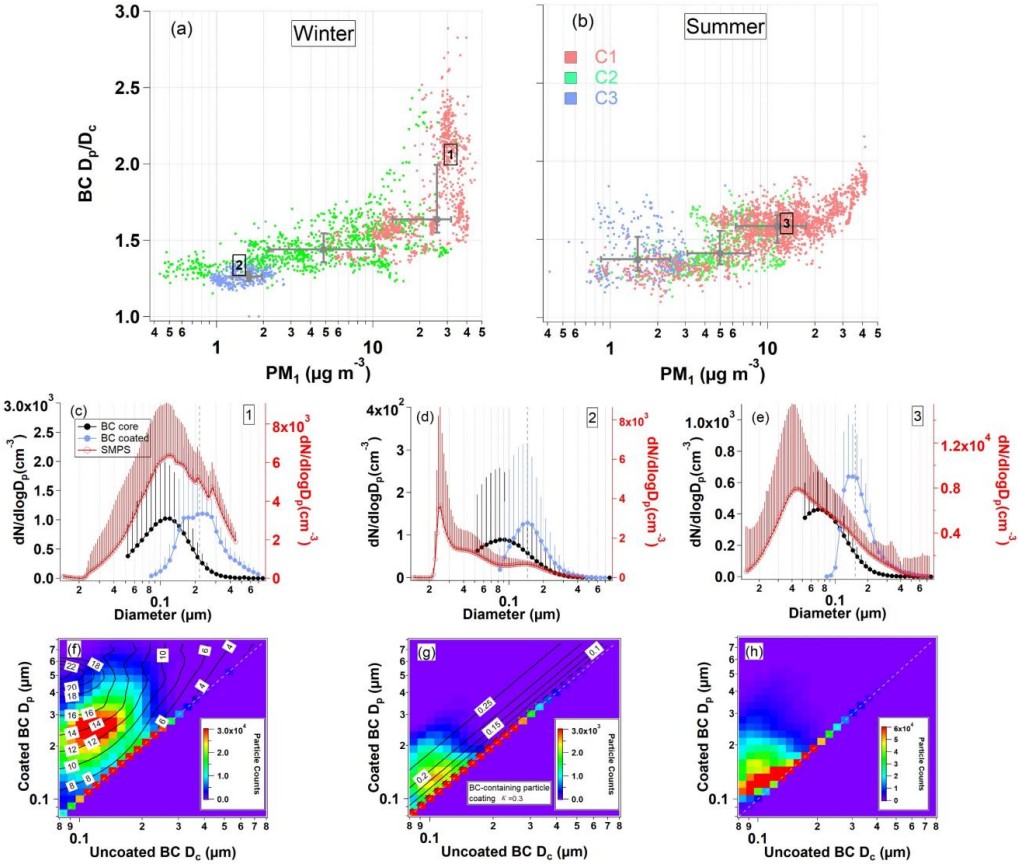

Fig. 5. Size-resolved mixing state of BC. The bulk relative coating thickness ($D_p/D_c$) as a function of $PM_1$ in
winter (a) and summer (b) for the three PBL types, with solid circles, whiskers denoting the median, 25th, 75th
percentiles respectively. Three typical periods, as marked as 1-3 in (a) and (b), are extracted for size
distribution analysis. (c),(d) and (e) are the corresponding number size distribution of all particles, uncoated
and coated BC for period 1-3 respectively. The bottom panels are coated BC diameter as a function of uncoated
BC diameter, colored by number density of single particle, where (f) and (g) are mapped with contour lines
numbered by the MAC and $\kappa_{BCc}$ respectively.

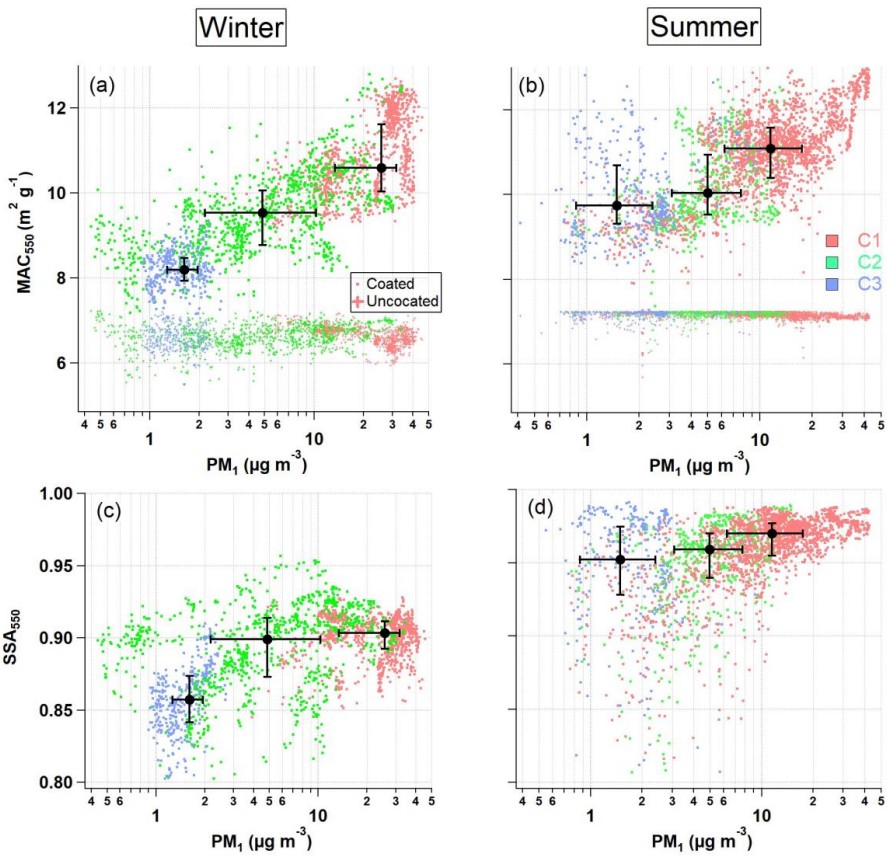

Fig. 6. Optical properties of BC for the three PBL types in both seasons. (a)(b) are mass absorption cross
section at $\lambda=550$ nm ($MAC_{550}$), with dot and plus markers denoting coated and uncoated BC respectively.
(c)(d) are single-scattering albedo at $\lambda=550$ nm ($SSA_{550}$). In each panel, the solid circles and whiskers denote
the median, 25th and 75th percentile respectively.



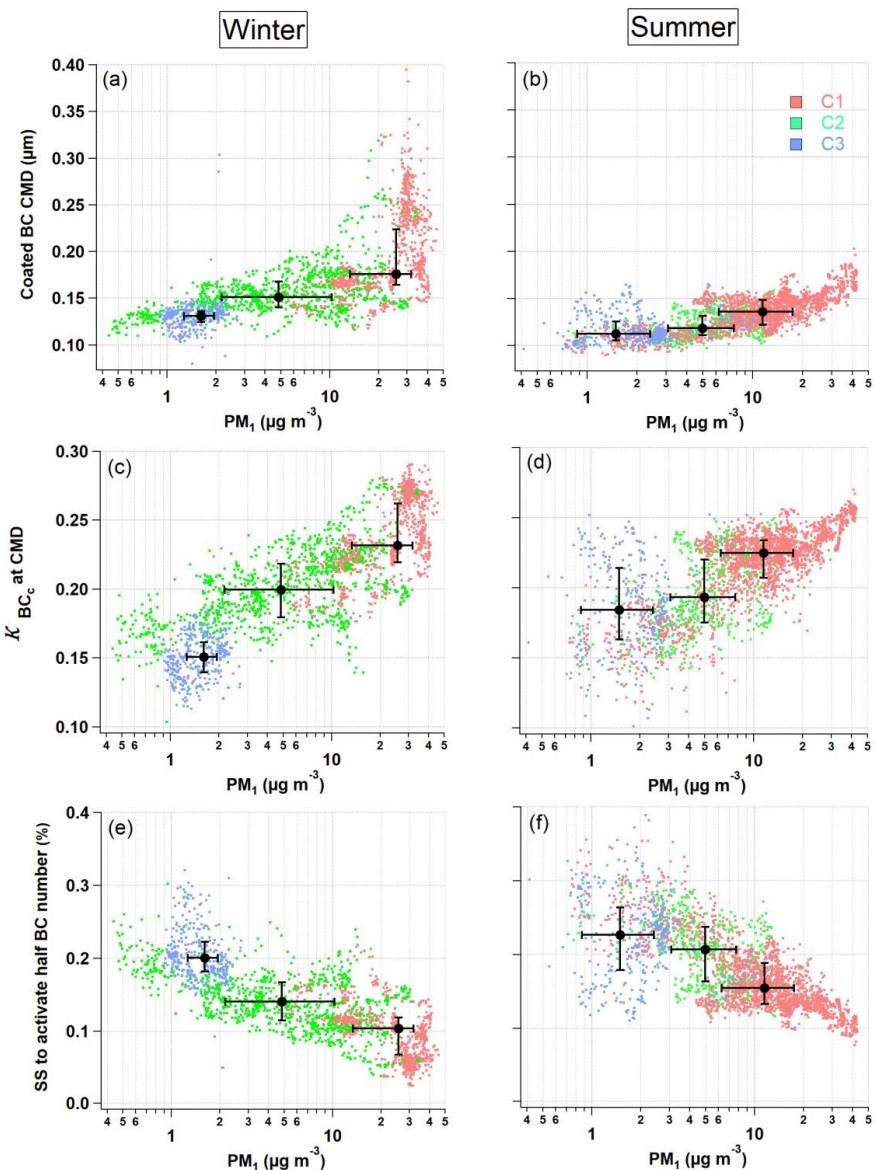

Fig. 7. Hygroscopic properties of BC for the three PBL types in both seasons. (a)(b) are the count median
diameter of coated BC; (c)(d) are the $\kappa_{BCc}$ assuming $\kappa_{coating}$ =0.3; (e)(f) are the supersaturation (SS) to activate
half of the BC number population. In each panel, the solid circles and whiskers denote the median, 25[th] and
75[th] percentile respectively.

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
