# Peer review of "Optical and hygroscopic properties of black carbon influenced by particle microphysics at the top of anthropogenically polluted boundary layer"

_Atmospheric Chemistry and Physics, 2020_

## Referee Comment (RC1) · Anonymous Referee #1 · 19 Oct 2020

General Comments: The manuscript (#acp-2020-762) entitled "Optical and hygroscopic properties of black carbon influenced by particle microphysics at the top of anthropogenically polluted boundary layer" by Ding et al presents detailed aerosol optical characteristics and hygroscopic nature of coated BC. The study seems to be interesting but the data length is too short. Also, the study lacks with the major implications. Thus, I recommend this paper for the "major revision", which needs to be revised as per the comments and suggestions given below. Specific Comments: 1. Line 13, Why summer MACBC is higher than the winter, if BC mass concentration is found to be relatively higher in winter than in summer? 2. In the "Introduction" and "Results and Discussion" sections, please include some comparisons and citations over various global regions

when analyzing the results of monthly and seasonal changes of BC optical characteristics, such as: Raju M. P. et al., Black carbon aerosols over a high altitude station, Mahabaleshwar: Radiative forcing and source apportionment. Atmospheric Pollution Research, 11, 1408-1417, 2020. Srivastava, A. K., S. Singh, P. Pant and U. C. Dumka: Characteristics of black carbon over Delhi and Manora Peak-a comparative study. Atmospheric Science Letters, 13, 223–230, 2012. Srivastava, A. K., D. S. Bisht, K. Ram, S. Tiwari and M. K. Srivastava: Characterization of carbonaceous aerosols over Delhi in Ganga basin: Seasonal variability and possible sources. Environmental Science and Pollution Research, 21, 8610-8619, 2014. Govardhan G., et al., Possible climatic implications of high-altitude black carbon emissions. Atmos. Chem. Phys., 17, 9623–9644, 2017. Vijayakumar S. Nair, S. Suresh Babu, K. Krishna Moorthy, Arun Kumar Sharma, Angela Marinoni & Ajai (2013) Black carbon aerosols over the Himalayas: direct and surface albedo forcing. Tellus B: Chemical and Physical Meteorology, 65:1, 19738, DOI: 10.3402/ tellusb.v65i0.19738. 3. Line 52-53, It is mentioned here that the study is done for the first time. However, several studies on BC microphysics have been published for mountain sites (few references are mentioned above). I would suggest to discuss and mention earlier studies. 4. The reported data length is too short. I would strongly suggest to include more possible data set to strongly support your findings. 5. Line 83, BC is highly absorbing in nature. How estimation of scattering cross-section of BC would be useful? 6. Line 159, why BC is high in winter as compared to summer when the station is found to be below the PBL? What are the major sources and how it reaches to the study site? 7. What are the uncertainties in the observed hygroscopic properties of BC as the pure BC is highly non-hygroscopic in nature? 8. It is not clear here that which type of particle coating is considered? How such coating or mixing state of particles with BC affects the overall atmospheric radiative characteristics? I would suggest to go through the paper by P. Srivastava et al., (Atmos. Res, 2018) and discuss a comparison accordingly. Srivastava, P., S. Dey, A. K. Srivastava, S. Singh, S. Tiwari: Most probable mixing state of aerosols in Delhi NCR, Northern India. Atmospheric Research, 200, 88-96, 2018.

9. In Figures 6 and 7, how mass concentration of PM1 was obtained? Please discuss in detail.

---

## Referee Comment (RC2) · Anonymous Referee #2 · 20 Oct 2020

Review of "Optical and hygroscopic properties of black carbon influenced by particle microphysics at the top of anthropogenically polluted boundary layer" by Ding et al.

This work presents the physical properties of particles containing black carbon measured during two short measurement campaigns (winter and summer 2019) on a mountain site in China. This site allows measurement at top of planetary boundary layer (PBL). For the full data set, the authors found and classified three types of PBL (according to back trajectories) from polluted regions to cleaner (and long range transportation) regions. The paper focuses mainly on comparing the black carbon containing particle physical properties of these three air masses, during winter and summer. The paper

reports the influence of coating on hygroscopic and optical properties of these particles. The data presented in the figures are easy and pleasant to read. The sentences are sometime long and therefore not easy to follow (for me), but after several reading I could understand all of them. Although the data set is short, it is enough to cover two distinct periods (winter vs summer) and provide a first hint of temporal variation of the PBL physical properties. I would recommend this paper for publication, after the following comments are clarified.

The paper would benefit to have more references and discussion about previous study. For example, in paragraph 38-48 the authors cite different studies on BC in the same regions, highlighting what was not measured. However, it would be beneficial for the paper, to also mention briefly, what are the main finding from these studies. Either here or in the discussion part.

line 6: "This study presents continuous measurements of detailed BC properties.." it would be fair to state that it presents x weeks of data during winter and y weeks of data during summer.

line 12: "we found enhanced BC mass absorption cross section (MACBC) for polluted PBL (up to 13 m2g-1 at $\lambda$=550nm), and summer had a higher MACBC than winter by 5%." for clarity of the sentence I suggest to change by for example: "...550nm), and that the PBL MACBC is higher by 5% during summer compare to winter.

line 158: "Consistent with the combined back-trajectory and emission analysis above, C1 had the highest BC for both seasons (1.0 $\pm$ 0.5 and 0.4$\pm$0.2 $\mu$g m-3 for winter and summer, respectively) and PM mass (23.8 $\pm$ 10.3 and 13.4$\pm$9.5 $\mu$g m-3). The concentration of BC mass was enhanced by a factor of 2.8 (1.7) higher than that in C2 for winter (summer), with winter having mass concentration frequently exceeding 1 $\mu$g m-3 The concentration of BC mass was enhanced by a factor of 2.8 (1.7) higher than that in C2 for winter (summer), with winter having mass concentration frequently exceeding 1 $\mu$g m-3."

This paragraph is difficult to follow. It would be easier if there is connection between sentences. For example here, it could written as follow:

"Consistent with the combined back-trajectory and emission analysis above, C1 had for both seasons the highest BC ($1.0 \pm 0.5$ and $0.4\pm0.2$ $\mu$g m-3 for winter and summer, respectively) and highest PM mass ($23.8 \pm 10.3$ and $13.4\pm9.5$ $\mu$g m-3). Compare to C1, for C2, the concentration of BC mass was enhanced by a factor of 2.8 (1.7) higher for winter (summer), with winter having mass concentration frequently exceeding 1 $\mu$g m-3. The concentration of BC mass was enhanced by a factor of 2.8 (1.7) higher than that in C2 for winter (summer), with winter having mass concentration frequently exceeding 1 $\mu$g m-3."

line 161: "It clearly shows" what (it) clearly show?

Fig1: in fig 1a, the sea could be also color, in blue for example. For non-local scientist, it would be easier to have a first idea of the region when the sea is also colored (the grey line for border between country or sea is not obvious for interpretation of the map). Fig1: Maybe a picture of the station been added as well?

―――――――――――――――

---

## Author Comment (AC1) · 10 Nov 2020

Dear Editor and Referees,

We thank for the important comments and suggestions on our manuscript. Our responses are in red and the changes we made in the manuscript are highlighted.

**Reviewer 1:**

General Comments: The manuscript (#acp-2020-762) entitled "Optical and hygroscopic properties of black carbon influenced by particle microphysics at the top of anthropogenically polluted boundary layer" by Ding et al presents detailed aerosol optical characteristics and hygroscopic nature of coated BC. The study seems to be interesting but the data length is too short. Also, the study lacks with the major implications. Thus, I recommend this paper for the "major revision", which needs to be revised as per the comments and suggestions given below.

We are thankful for the constructive comments from the reviewer. We have adopted the reviewer's suggestions and revised the conclusion and introduction section to extend the implications of our study. We have conducted the experiments with intensive measurements lasting for one month for each representative season. The one-month intensive measurement tends to be a common approach for most of the ground-based campaigns (Liu et al., 2014;Holzinger et al., 2013;Adachi and Buseck, 2013;Yang et al., 2018;Metcalf et al., 2012), but to conduct a longer experiment may not add more or influence on our main conclusions derived here.

Specific comments:

 Line 13, Why summer MACBC is higher than the winter, if BC mass concentration is found to be relatively higher in winter than in summer? MACBC is the absorption per unit BC mass, which is determined by the BC core

size and coatings. BC in summer had smaller core size than in winter thus larger  $MAC_{BC}$ . We have clarified this point in the revision:

"The overall higher absorption efficiency of BC in summer (especially in diluted PBL) was due to the smaller core size, leading to a higher baseline MAC for uncoated BC by 10% than winter. This effect prevailed the even lower coating amount and associated absorption enhancement in summer (Fig. 5b)." Line 254-256

2. In the "Introduction" and "Results and Discussion" sections, please include some comparisons and citations over various global regions when analyzing the results of monthly and seasonal changes of BC optical characteristics, such as: Raju M. P. et al., Black carbon aerosols over a high altitude station, Mahabaleshwar: Radiative forcing and source apportionment. Atmospheric Pollution Research, 11, 1408-1417, 2020. Srivastava, A. K., S. Singh, P. Pant and U. C. Dumka: Characteristics of black carbon over Delhi and Manora Peak-a comparative study. Atmospheric Science Letters, 13, 223–230, 2012. Srivastava, A. K., D. S. Bisht, K. Ram, S. Tiwari and M. K. Srivastava: Characterization of carbonaceous aerosols over Delhi in Ganga basin: Seasonal variability and possible sources. Environmental Science and Pollution Research, 21, 8610-8619, 2014. Govardhan G., et al., Possible climatic

implications of high-altitude black carbon emissions. Atmos. Chem. Phys., 17, 9623–9644, 2017. Vijayakumar S. Nair, S. Suresh Babu, K. Krishna Moorthy, Arun Kumar Sharma, Angela Marinoni & Ajai (2013) Black carbon aerosols over the Himalayas: direct and surface albedo forcing. Tellus B: Chemical and Physical Meteorology, 65:1, 19738, DOI: 10.3402/ tellusb.v65i0.19738.

We have updated the references and added more comparisons with other literatures suggested by the reviewer in our revised manuscript:

"while the heating can promote the convection above (Mcfarquhar and Wang, 2006;Rudich et al., 2003), e.g. BC was found to account for 43% of the total aerosol radiative forcing at the atmosphere in south Asia due to heating. (Raju et al., 2020). Line 32-33

In winter C2 showed clear diurnal variation of BC and PM1 concentration with significant enhancement in the daytime (Fig. 4b and e) when fully developed PBL (Fig. 2d), with the diurnal pattern consistent with a previous study conducted in the Indian Himalayan foothills (Srivastava et al., 2012b). Line 158-159

C3 had the lowest pollution level among all air mass clusters with a lower BC  $(0.09\pm0.03 \ \mu g \ m^{-3})$  and PM1 mass  $(1.8\pm0.4 \ \mu g \ m^{-3})$  than C2 by a factor of 2-4, within a similar range of a previous study in western and central Himalayas (Nair et al., 2013).

Line 180-181

Rather than being subject to significant scavenging processes of low-level emissions, BC transported to the lower free troposphere may be transported to a wider region (Weinzierl, 2008; Yang et al., 2018; Govardhan et al., 2017), exerting regional direct and indirect radiative impacts." Line 327

3. Line 52-53, It is mentioned here that the study is done for the first time. However, several studies on BC microphysics have been published for mountain sites (few references are mentioned above). I would suggest to discuss and mention earlier studies.

We have updated the references and discussions per reviewer's suggestion, as answered above.

4. The reported data length is too short. I would strongly suggest to include more possible data set to strongly support your findings.

We thank the reviewer for this suggestion. The main objective of this study is to investigate the microphysical properties of BC during two contrast seasons, rather than long-term seasonal variability. This requires intensive application of instrumentations on the site and the long-term application was not available for our dataset. The one-month intensive experiments have been widely conducted during a number of international campaigns (Liu et al., 2014;Holzinger et al., 2013;Adachi and Buseck, 2013;Yang et al., 2018;Metcalf et al., 2012). Based on our data set, the

information of science had been conveyed, but longer data may not add more or influence on our conclusions derived here.

- Line 83, BC is highly absorbing in nature. How estimation of scattering crosssection of BC would be useful?
  We have been using total aerosol to estimate the scattering, as BC only took a small mass fraction (<10%), which is unimportant in contributing the total scattering of aerosol ensemble.
- 6. Line 159, why BC is high in winter as compared to summer when the station is found to be below the PBL? What are the major sources and how it reaches to the study site?

The reason for higher BC mass in winter is the higher surface emission and lower scavenging in winter, though the PBL was not as developed as in summer. The transport mechanism is detailed in section 2 and section 4.1. To clarify, we have added some linkage in that part the reviewer mentioned.

"The enhanced concentration from midday in summer was also observed (on the mean, Fig. 4a) but not as pronounced as winter, probably due to some wet removal during the daytime vertical transport from the surface, given the high RH in the summer.

The less concentration in summer mountain may also result from the lower surface emission in the season."

Line 166-168, Line 177-178.

7. What are the uncertainties in the observed hygroscopic properties of BC as the pure BC is highly non-hygroscopic in nature?

The hygroscopicity parameter of BC in this study was calculated based on the Zdanovskii–Stokes–Robinson (ZSR) rule with the assumption of  $\kappa_{rBC}=0$  and  $\kappa_{coating}=0.3$ , by measuring the size-resolved mixing state of BC. The hygroscopic properties of BC were hereby determined by the hygroscopicity of coatings and the core of BC. This is in detail explained in section 3.

8. It is not clear here that which type of particle coating is considered? How such coating or mixing state of particles with BC affects the overall atmospheric radiative characteristics? I would suggest to go through the paper by P. Srivastava et al., (Atmos. Res, 2018) and discuss a comparison accordingly. Srivastava, P., S. Dey, A. K. Srivastava, S. Singh, S. Tiwari: Most probable mixing state of aerosols in Delhi NCR, Northern India. Atmospheric Research, 200, 88-96, 2018.

We thank for this comment from the reviewer. We only quantify the abundance of the coatings, and to what extent BC particles were coated at different sizes. The coating composition was not measured during this study. We have updated the discussions and incorporated the information from the literatures the reviewer suggested.

"The entire particle size of BC-containing particle (Dp) including coatings is determined by matching the measured scattering cross section with the modelled one based on core-shell assumption using a Mie lookup table (Liu et al., 2014;Taylor et al., 2015). This assumption is mostly valid for BC with thick

**coatings (Liu et al., 2017), but not considering the scenario in which BC may be attached to dust particle (Srivastava et al., 2018), given there was no dust event observed in this study." Line 94-96.**

9. In Figures 6 and 7, how mass concentration of PM1 was obtained? Please discuss in detail.

This is clarified in revised figure caption.

"Fig. 3. Time series of BC mass (a)(d), PM1 derived from the SMPS size distribution (b)(e), BC mass fraction (c)(f) at both seasons, shaded by the periods in identified three clusters, by red, green and blue corresponding to cluster 1 (C1), cluster 2 (C2), cluster 3 (C3) respectively. Frequency histograms of BC mass (g)(h), PM1 (i)(j), and BC mass fraction (k)(l) for each cluster in both seasons." Line 359.

**Reviewer 2:**

This work presents the physical properties of particles containing black carbon measured during two short measurement campaigns (winter and summer 2019) on a mountain site in China. This site allows measurement at top of planetary boundary layer (PBL). For the full data set, the authors found and classified three types of PBL (according to back trajectories) from polluted regions to cleaner (and long range transportation) regions. The paper focuses mainly on comparing the black carbon containing particle physical properties of these three air masses, during winter and summer. The paper reports the influence of coating on hygroscopic and optical properties of these particles. The data presented in the figures are easy and pleasant to read. The sentences are sometime long and therefore not easy to follow (for me), but after several reading I could understand all of them. Although the data set is short, it is enough to cover two distinct periods (winter vs summer) and provide a first hint of temporal variation of the PBL physical properties. I would recommend this paper for publication, after the following comments are clarified.

We are thankful for the comprehensive summary of our study and the positive comments from the reviewer.

1. The paper would benefit to have more references and discussion about previous study. For example, in paragraph 38-48 the authors cite different studies on BC in the same regions, highlighting what was not measured. However, it would be beneficial for the paper, to also mention briefly, what are the main finding from these studies. Either here or in the discussion part.

We have added more detailed discussion on the references in line 41-45.

"Ground measurements of BC have been intensively conducted over the polluted North China Plain (NCP) region in last decades (Han et al., 2009;Cheng et al., 2011;Ji et al., 2018;Liu et al., 2019a), e.g. the seasonal variation of elemental carbon in urban Beijing during 2005-2006 and 2012-2013 (Han et al., 2009;Ji et al., 2018), the size-resolved mixing state of BC and aging mechanism at a suburban site in NCP (Cheng et al., 2011), and the contrast physical properties in urban Beijing between winter and summer (Liu et al., 2019a), were investigated in these studies." Line 41-45.

2. line 6: "This study presents continuous measurements of detailed BC properties.." it would be fair to state that it presents x weeks of data during winter and y weeks of data during summer.

This sentence has been revised as:

"This study presents continuous measurements of detailed BC properties at a mountain site locating at the top of polluted PBL over the North China Plain, during seasons (3 and 4 weeks of data during winter and summer, respectively) with contrast emission structure and meteorology." Line 5-8.

line 12: "we found enhanced BC mass absorption cross section (MACBC) for polluted PBL (up to 13 m2g-1 at λ=550nm), and summer had a higher MACBC than winter by 5%." for claritity of the sentence I suggest to change by for example:
...550nm), and that the PBL MACBC is higher by 5% during summer compare

**to winter.**

This sentence has been revised as:

"By resolving the detailed particle size-resolved mixing state of BC in optical and hygroscopic models, we found enhanced BC mass absorption cross section (MACBC) for polluted PBL (up to 13 m2g-1 at  $\lambda$ =550 nm), which was 5% higher during summer than winter due to smaller BC core size." Line 11-14.

4. line 158: "Consistent with the combined back-trajectory and emission analysis above, C1 had the highest BC for both seasons  $(1.0 \pm 0.5 \text{ and } 0.4 \pm 0.2 \text{ } \mu\text{g m-3} \text{ for}$ winter and summer, respectively) and PM mass  $(23.8 \pm 10.3 \text{ and } 13.4 \pm 9.5 \mu \text{g m-3})$ . The concentration of BC mass was enhanced by a factor of 2.8 (1.7) higher than that in C2 for winter (summer), with winter having mass concentration frequently exceeding 1 µg m-3 The concentration of BC mass was enhanced by a factor of 2.8 (1.7) higher than that in C2 for winter (summer), with winter having mass concentration frequently exceeding 1 µg m-3." This paragraph is difficult to follow. It would be easier if there is connection between sentences. For example here, it could written as follow: "Consistent with the combined back-trajectory and emission analysis above, C1 had for both seasons the highest BC ( $1.0 \pm 0.5$  and  $0.4\pm0.2 \,\mu g$  m-3 for winter and summer, respectively) and highest PM mass (23.8  $\pm$ 10.3 and 13.4±9.5 µg m-3). Compare to C1, for C2, the concentration of BC mass was enhanced by a factor of 2.8 (1.7) higher for winter (summer), with winter having mass concentration frequently exceeding 1 µg m-3. The concentration of BC mass was enhanced by a factor of 2.8 (1.7) higher than that in C2 for winter (summer), with winter having mass concentration frequently exceeding 1 µg m-3." We are thankful for the very careful reading and comments from the reviewer, and all the comments above have been revised accordingly :

"Consistent with the combined back-trajectory and emission analysis above, C1 had for both seasons the highest BC ( $1.0 \pm 0.5$  and  $0.4 \pm 0.2 \ \mu g \ m^{-3}$  for winter and summer, respectively) and highest PM mass ( $23.8 \pm 10.3$  and  $13.4 \pm 9.5 \ \mu g \ m^{-3}$ ). Compared to C1, for C2, the concentration of BC mass was enhanced by a factor of 2.8 (1.7) higher than that in C2 for winter (summer), with winter having mass concentration frequently exceeding 1  $\mu g \ m^{-3}$ ."

Line 169-173.

- line 161: "It clearly shows" what (it) clearly show? This sentence has been changed as "C1 clearly shows.....". line 173
- 6. Fig1: in fig 1a, the sea could be also color, in blue for example. For non-local scientist, it would be easier to have a first idea of the region when the sea is also colored (the grey line for border between country or sea is not obvious for interpretation of the map).

Fig. 1 has been revised for clearer demonstration.

 Fig1: Maybe a picture of the station been added as well? The photo of the mountain station has been added in Fig1. (d).

**References**

Adachi, K., and Buseck, P. R.: Changes of ns-soot mixing states and shapes in an urban area during CalNex, Journal of Geophysical Research: Atmospheres, 118, 3723-3730, https://doi.org/10.1002/jgrd.50321, 2013.

Holzinger, R., Goldstein, A. H., Hayes, P. L., Jimenez, J. L., and Timkovsky, J.: Chemical evolution of organic aerosol in Los Angeles during the CalNex 2010 study, Atmos. Chem. Phys., 13, 10125-10141, 10.5194/acp-13-10125-2013, 2013.

Liu, D., Allan, J. D., Young, D. E., Coe, H., Beddows, D., Fleming, Z. L., Flynn, M. J., Gallagher, M. W., Harrison, R. M., Lee, J., Prevot, A. S. H., Taylor, J. W., Yin, J., Williams, P. I., and Zotter, P.: Size distribution, mixing state and source apportionment of black carbon aerosol in London during wintertime, Atmospheric Chemistry and Physics, 14, 10061-10084, 10.5194/acp-14-10061-2014, 2014.

Liu, D., Whitehead, J. D., Alfarra, M. R., Reyesvillegas, E., Spracklen, D. V., Reddington, C. L., Kong, S., Williams, P. I., Ting, Y., and Haslett, S. L.: Black-carbon absorption enhancement in the atmosphere determined by particle mixing state, Nature Geoscience, 10, 184-188, 2017.

Mcfarquhar, G. M., and Wang, H.: Effects of aerosols on trade wind cumuli over the Indian Ocean: Model simulations, Quarterly Journal of the Royal Meteorological Society, 132, 821-843, 2006.

Metcalf, A. R., Craven, J. S., Ensberg, J. J., Brioude, J., Angevine, W., Sorooshian, A., Duong, H. T., Jonsson, H. H., Flagan, R. C., and Seinfeld, J. H.: Black carbon aerosol over the Los Angeles Basin during CalNex, Journal of Geophysical Research: Atmospheres, 117, https://doi.org/10.1029/2011JD017255, 2012.

Raju, M. P., Safai, P. D., Sonbawne, S. M., Buchunde, P. S., Pandithurai, G., and Dani, K. K.: Black carbon aerosols over a high altitude station, Mahabaleshwar: Radiative forcing and source apportionment, Atmospheric Pollution Research, 11, 1408-1417, https://doi.org/10.1016/j.apr.2020.05.024, 2020.

Rudich, Y., Sagi, A., and Rosenfeld, D.: Influence of the Kuwait oil fires plume (1991) on the microphysical development of clouds, Journal of Geophysical Research, 108, 10.1029/2003JD003472, 2003.

Yang, W., Zhang, Y., Wang, X., Li, S., Zhu, M., Yu, Q., Li, G., Huang, Z., Zhang, H., Wu, Z., Song, W., Tan, J., and Shao, M.: Volatile organic compounds at a rural site in Beijing: Influence of temporary emission control and wintertime heating, Atmospheric Chemistry and Physics Discussions, 1-55, 10.5194/acp-2018-29, 2018.

---

## Author Response (AR2)

Dear Editor and Referees,

We thank for the important comments and suggestions on our manuscript. Our responses are in red and the changes we made in the manuscript are highlighted.

Reviewer 1:

I thank the author for their clear responses which have remove my previous concern as well as the concerns raise by reviewer 1 (to my opinion). For me the manuscript is ready to be published as it is with only one small technical correction.

Figure 1 a, the sea now seems to be all other China. It would be easier if the author could change the elevation color to clarify this. For example by taking a color scale using color from Green at 0 m asl up to brown in the mountain and only keep blue color for the sea/river.

We thank the reviewer for this suggestion, and we have modified Fig. 1a according to the reviewer's advice.

Fig. 1. Experimental site descriptions. (a) The location of the experimental site and central Being, marked with black and red star respectively, where the color bar denotes the terrian height and blue denotes the sea. (b) the monthly BC emission inventory in China (Li et al., 2017). (c) Schematic illustration for different types of PBL defined in this study. (d) Photo of the mountain station.

Reviewer 2:

The authors have revised the manuscript thoroughly and I would suggest accepting it for publication after addressing the below minor concern.

To extend the potential radiative implication of the study, I would suggest to substantiate your results with the recently published article by Srivastava et al., (Atm. Environ., 2020).

Srivastava, A. K., et al. Implications of different aerosol species to direct radiative forcing and atmospheric heating rate. Atmospheric Environment, 241, 117820, 2020.

We thank the reviewer for this suggestion, and we have added this in our revised manuscript.

The single-scattering albedo at $\lambda$=550nm ($SSA_{550}$) in winter was systematically lower for all PBL types (Fig. 6 c-d and Fig. S4), lowered by 0.06, 0.05 and 0.08 than summer for diluted, neutral and polluted PBL, respectively. The decreased $SSA_{550}$ was in line with the increased BC mass faction (Fig. 3), and also influenced by the absorbing efficiency, as it was reported that the SSA of elemental carbon (EC) particles was as low as 0.25 (Srivastava et al., 2020). Line 264-265

Li, M., Zhang, Q., Kurokawa, J.-i., Woo, J.-H., He, K., Lu, Z., Ohara, T., Song, Y., Streets, D. G., Carmichael, G. R., Cheng, Y., Hong, C., Huo, H., Jiang, X., Kang, S., Liu, F., Su, H., and Zheng, B.: MIX: a mosaic Asian anthropogenic emission inventory under the international collaboration framework of the MICS-Asia and HTAP, Atmospheric Chemistry and Physics, 17, 935-963, 10.5194/acp-17-935-2017, 2017.

Srivastava, A. K., Mehrotra, B. J., Singh, A., Singh, V., Bisht, D. S., Tiwari, S., and Srivastava, M. K.: Implications of different aerosol species to direct radiative forcing and atmospheric heating rate, Atmospheric Environment, 241, 117820, https://doi.org/10.1016/j.atmosenv.2020.117820, 2020.